# Federated Learning Based Fault Diagnosis Driven by Intra-Client Imbalance Degree

**DOI:** 10.3390/e25040606

**Published:** 2023-04-03

**Authors:** Funa Zhou, Yi Yang, Chaoge Wang, Xiong Hu

**Affiliations:** School of Logistic Engineering, Shanghai Maritime University, Shanghai 201306, China

**Keywords:** fault diagnosis, federated learning, imbalance mode mismatch, intra-client imbalance degree, joint optimization

## Abstract

Federated learning is an effective means to combine model information from different clients to achieve joint optimization when the model of a single client is insufficient. In the case when there is an inter-client data imbalance, it is significant to design an imbalanced federation aggregation strategy to aggregate model information so that each client can benefit from the federation global model. However, the existing method has failed to achieve an efficient federation strategy in the case when there is an imbalance mode mismatch between clients. This paper aims to design a federated learning method guided by intra-client imbalance degree to ensure that each client can receive the maximum benefit from the federation model. The degree of intra-client imbalance, measured by gain of a class-by-class model update on the federation model based on a small balanced dataset, is used to guide the designing of federation strategy. An experimental validation for the benchmark dataset of rolling bearing shows that a 23.33% improvement of fault diagnosis accuracy can be achieved in the case when the degree of imbalance mode mismatch between clients is prominent.

## 1. Introduction

As a key component of a motor, fault diagnosis research on rolling bearings plays an important role in ensuring the safe and stable operation of the motor [1,2,3]. As an effective data-driven fault diagnosis approach, deep learning is not limited by a precise physical model or adequate expert knowledge and can automatically extract fault features from raw data [4,5,6]. Therefore, the fault diagnosis methods based on deep learning have received widespread attention.

Deep learning is a powerful feature extraction tool since it can approximate complex functions by using layer-by-layer feature extraction [7]. According to the different network structures used, fault diagnosis based on deep learning can be divided into four categories: deep belief network (DBN), convolutional neural network (CNN), recurrent neural network (RNN), and stacked autoencoder (SAE) [8,9,10,11].

The effectiveness of deep learning fault diagnosis relies on massive balanced data. However, rolling bearings usually operate under normal conditions and fault samples are difficult to obtain, so the data collected are often imbalanced. The data imbalance problem reduces the effectiveness of deep learning fault diagnosis [12,13]. To reduce the impact of imbalanced data on model performance of deep learning, data-level and algorithm-level approaches are proposed [14,15,16,17,18]. Among the data-level methods, oversampling and undersampling techniques are used to construct a balanced dataset [14,15,16]. However, oversampling can produce duplicate information, while undersampling can lead to a loss of information. To avoid generating duplicate information, Zhu et al. [17] proposed a fault diagnosis method that combines generative adversarial networks and convolutional neural networks to generate fake samples with real sample characteristics to balance the original dataset. However, the reliability of generated samples cannot be guaranteed when the quality of the real samples is poor. As the algorithm-level method, cost-sensitive learning can extract fault features directly from the imbalanced data by assigning class weights to increase the dominance of minority class samples in model training [18]. However, it is difficult to build an effective deep learning fault diagnosis model when the sample size of a single client is insufficient. Therefore, it is important to develop an effective fault diagnosis method jointly with different client models.

Federated learning is a distributed machine learning algorithm that can realize the joint optimization of individual clients to obtain a robust global fault diagnosis model under the premise of ensuring data privacy [19,20,21]. The accuracy of the federated learning fault diagnosis relies on balanced data from each client. However, clients often suffer from data imbalance problems in actual industrial sites and the data samples on each client are usually not independent or identically distributed, which leads to a serious statistical heterogeneity and reduces the effectiveness of federated learning [22].

Dealing with the data imbalance problem in federated learning can improve the fault diagnosis accuracy. Zhao et al. [23] pre-trained the initialized global model using uniformly distributed global shared data stored in the federated center. In addition, part of the global shared data is sent to all clients to balance the training data of each client. However, it is difficult to decide which part of the data to share and the proportion of the shared data. Duan et al. [24] proposed a data augmentation approach to balance the dataset by generating a minority class of samples through linear interpolation between nearest neighbors. However, this approach tends to lead to the overfitting of the local model due to the generation of duplication information. Xin et al. [25] introduced a generative adversarial network in federated learning to generate missing class samples for each client. However, the reliability of generated samples cannot be guaranteed when the quality of true samples is poor. McMahan et al. [26] proposed a traditional federated average algorithm. The federation aggregation weight of each client is determined by the number of training samples, and then the global model parameters are obtained by aggregating the local model parameters of each client. To better deal with the imbalance issue in federated learning, Ma et al. [27] proposed a hierarchical federation aggregation strategy, the aggregation weight of the feature extraction layer of each client model is determined by the number of training samples, and the aggregation weight of each class in the classification layer is determined by the number of class samples. To avoid keeping the aggregation weights of each client fixed in each round of federation communication, some researchers have designed a dynamic federation aggregation strategy. Geng et al. [28] proposed an improved federation aggregation strategy by adding the model classification metrics F1-score to the federation aggregation strategy and assigning greater aggregation weights to clients with better classification results. Xiao et al. [29] proposed an accuracy-based federation aggregation strategy, which determined the aggregation weight of each client based on the accuracy of each client model on the public validation dataset. Wang et al. [30] proposed an imbalanced federated learning method incorporating cost-sensitive learning, the model gain after updating the federated aggregation model separately by category using the balanced dataset of the federated center to measure the degree of imbalance for each client to enable local cost-sensitive learning for each client. However, the degree of imbalance in the federated center data on the aggregation model is not representative of the actual imbalance degree of each client, which will inevitably affect the effectiveness of federated learning.

In this work, we designed a mechanism to accurately measure the degree of imbalance for each client, using the model gain when updating the balanced small sample model with the imbalance dataset as the unique imbalance degree for each client, and developed a federation strategy between clients through cost-sensitive learning. Using the proposed method, we can better solve the problem of identifying the optimal federation strategy construction when there is an imbalance mode mismatch between clients. The contribution of this paper is as follows: A federated learning framework driven by intra-client imbalance degree was designed in this paper to establish a federation strategy for the case where there is an imbalance mode mismatch between clients.The degree of intra-client imbalance was used to guide the design of the federation strategy related to cost-sensitivity. Using the imbalance data of the local client, an inter-class imbalance degree for each client was computed by the gain of a class-by-class model update on the federation model of the small balanced dataset.In the case where there is a significant mismatch of imbalance mode between clients, the federated learning-based fault diagnosis proposed in this paper can well overcome the problem arisen by both intra-client imbalance and inter-client imbalance to ensure the accuracy of fault diagnosis for each client.

## 2. Related Theories

### 2.1. Deep Neural Network Based on Stack Autoencoder

An autoencoder (AE) is an unsupervised neural network model that can learn the hidden features of the input data by encoding and reconstructing the original input data by decoding [31]. The feature extraction power of a single AE can be increased by stacking multiple AEs to extract high-dimensional features for feeding into the classifier, thus forming a deep neural network (DNN) to undertake a fault diagnosis. Finally, global fine-tuning is performed using the labelled input data [32]. The DNN network structure is shown in Figure 1.

### 2.2. Federated Learning

Federated learning is a distributed machine learning algorithm that enables the joint training of multiple clients without sharing data directly. The schematic of federated learning is shown in Figure 2. In each round of federation communications, each client downloads global model parameters to train using private labeled data and then uploads the updated local model parameters to the federated center. The federated center updates global model parameters by aggregating the local model parameters of each client. When the number of federations is reached, each client uses a test dataset to produce a fault diagnosis.

## 3. A Federated Learning Method Driven by Intra-Client Imbalance Degree

Designing a federation strategy that considers the degree of intra-client imbalance is particularly important for improving the performance of federated learning. This is because the degree of intra-client imbalance can measure the contribution of each class sample to the federation model, and this can be used to guide the design of federation strategy related to cost-sensitivity. Therefore, this section outlines a federated learning method driven by intra-client imbalance degree (Fed_ICID) to measure the unique inter-class imbalance degree of each client as the contribution to the federation model. Additionally, the federation strategy between clients related to cost-sensitivity is established by using the degree of intra-client imbalance to ensure that the local client can benefit from the federation model to the maximum extent and achieve fault diagnosis accuracy. The specific steps are as follows.

### 3.1. A Federated Learning Framework Driven by Intra-Client Imbalance Degree

When there is an imbalance mode mismatch between clients, the degree of intra-client imbalance can be used to drive the design of local model training and the federation strategy related to cost-sensitivity so that each client can receive the maximum benefit from the federation model and improve the fault diagnosis accuracy. Therefore, we designed a federated learning framework driven by the degree of intra-client imbalance to weaken the negative effects of imbalanced data on federated learning, as shown in Figure 3.

Since the data on each client are imbalanced, the local model training of each client directly on the global model will affect the federated learning. Therefore, the balanced data with a small sample size for each client are constructed by means of a sample balancing. The imbalanced data of the local client are used to compute the inter-class imbalance degree by the gain of a class-by-class model update on the federation aggregation model using balanced data. The inter-class imbalance degree of each client can be used for local cost-sensitive learning. By aggregating the cost-sensitive loss value of each client, the federated center loss function is constructed to drive the joint learning of local model parameters and aggregation weights for each client, thereby ensuring that each client can achieve the maximum benefit from the federation global model.

### 3.2. Detailed Steps for Fed_ICID Method

The degree of intra-client imbalance aims to measure the contribution degree of each client to the federation model, which can be used to guide the design of the federation strategy related to cost-sensitivity. This is because the degree of the intra-client imbalance can accurately measure the contribution of each class sample to the federation model, which is important for improving the effectiveness of federated learning. The detailed steps of the proposed federated learning fault diagnosis method driven by intra-client imbalance degree are shown below. The block diagram of the proposed Fed_ICID method is shown in Figure 4.
**Step 1:** **Define the training dataset for each client**

$X_{1},\ldots ,X_{k},\ldots ,X_{K}$ are the imbalanced dataset of Client 1,…,Client k,…,Client K, respectively, and there is imbalance mode mismatch between clients. There are C classes of fault types included in each dataset $X_{k}$ with a limited size of labeled training samples. It is assumed that $X_{1B},\ldots ,X_{kB},\ldots ,X_{KB}$ are the corresponding balanced dataset determined by the minority class of the imbalance dataset.
**Step 2:** **Establish the federation model based on the small balanced dataset**

The federation model established by the small balanced dataset of each client can be formulated as: (1)$$W_{g,s+1,B}=\sum_{k=1}^{K}p_{k,s,B}W_{k,s,B}$$
where $W_{g,s+1,B}$ denotes the federation model parameters using the small balanced dataset of all clients. $W_{k,s,B}$ denotes the model parameters of the kth client in the sth round of federation. The corresponding aggregation weights $p_{k,s,B}$ are determined by the step 4.
**Step 3:** **Measure the intra-client imbalance degree by gain of the class-by-class model update of the federation model**

Using the imbalanced data of the local client, the inter-class imbalance degree for each client can be computed by the gain of the class-by-class model update of the federation model $W_{g,s+1,B}$, as shown in Equation (2).
(2)$$\alpha_{k,s,c}=\frac{\sum_{c=1}^{C}(\Delta W_{k,s,c})-\Delta W_{k,s,c}}{(C-1)\Delta W_{k,s,c}}$$
where $\Delta W_{k,s,c}$ denotes the model gain using samples of the cth class for the kth client in the sth round of federations, $\alpha_{k,s,c}$ denotes the corresponding weight of cth class for the kth client. $\Delta W_{k,s,c}$ can be obtained by Equation (3).
(3)$$\Delta W_{k,s,c}=W_{k,s,c}-W_{g,s+1,B}$$
$W_{k,s,c}$ denotes the updated model parameters trained by samples of the cth class for the kth client. The training process for the class-by-class update of $W_{g,s+1,B}$ is driven by the loss function formulated by Equation (4).
(4)$$loss_{k,s,c}=\frac{1}{2L_{k,c}}\sum_{l=1}^{L_{k,c}}{\left(y_{k,c,l}-{\tilde{y}}_{k,c,l}\right)}^{2}$$
where $y_{k,c,l}$ denotes the true label of the lth sample of the cth class for the kth client, and ${\tilde{y}}_{k,c,l}$ denotes the corresponding predicted label of network model. $L_{k,c}$ denotes the sample size of the cth class for the kth client. The update process of local model parameters $W_{k,s,c}$ can be obtained by Equation (5) using the gradient descent optimization algorithm.
(5)$$W_{k,s,c}=W_{g,s+1,B}-lr\nabla loss_{k,s,c}(W_{g,s+1,B})$$
where lr denotes the learning rate. Then the cost-sensitive loss function $loss_{k,s}$ using the imbalanced dataset $X_{k}$ of kth client in the sth round of federations on $W_{g,s+1,B}$ can be obtained by Equation (6).
(6)$$loss_{k,s}=\frac{1}{2L_{k}}\sum_{l=1}^{L_{k,c}}\sum_{c=1}^{C}(1+\alpha_{k,s,c}){\left(y_{k,c,l}-{\tilde{y}}_{k,c,l}\right)}^{2}$$
where $L_{k}$ denotes the training sample size of the kth client. After the kth client completes local training, the corresponding updated local model parameters $W_{k,s}$ are uploaded to the federated center, and the global model parameters $W_{g,s+1}$ are obtained by step 4.
**Step 4:** **Federation aggregation strategy driven by intra-client imbalance degree**

The intra-client imbalance degree will inevitably affect the efficiency of the federation. A federation aggregation strategy related to cost-sensitivity driven by intra-client imbalance degree is proposed in this part, as shown in Equation (7).
(7)$$loss_{g,s}^{}=\sum_{k=1}^{K}p_{k,s}^{}loss_{k,s}^{}$$
where $p_{k,s}^{}$ denotes the aggregation weights of the kth client in the sth round of federations; it can be learned by minimizing the loss function designed in Equation (7). $loss_{k,s}^{}$ denotes the cost-sensitive loss function used in the federation model; it can be computed by forward propagation of the network model for the kth client, as shown in Equation (8):(8)$$loss_{k,s}=\frac{1}{2L_{k}}\sum_{l=1}^{L_{k,c}}\sum_{c=1}^{C}(1+\alpha_{k,s,c}){\left(y_{k,c,l}-W_{k,s}X_{k,c,l}\right)}^{2}$$
where $X_{k,c,l}$ denotes the lth sample of cth class for kth client. The local model defined in Equation (8) and federation model defined in Equation (7) can be jointly optimized. The global loss function can be determined by substituting Equation (8) into Equation (7), as shown in Equation (9):(9)$$loss_{g,s}^{}=\frac{1}{2L_{k}}\sum_{k=1}^{K}\sum_{l=1}^{L_{k,c}}\sum_{c=1}^{C}p_{k,s}^{}(1+\alpha_{k,s,c}){\left(y_{k,c,l}-W_{k,s}X_{k,c,l}\right)}^{2}$$

The detailed update process of joint optimization for local model parameters and aggregation weights can be computed in Equations (10) and (11), respectively.
(10)$$W_{k,s}^{}=W_{k,s}^{}-lr\nabla W_{k,s}^{}$$
(11)$$p_{k,s}^{}=p_{k,s}^{}-lr\nabla p_{k,s}^{}$$
where $\nabla W_{k,s}^{}$ and $\nabla p_{k,s}^{}$ can be determined by Equations (12) and (13), respectively.
(12)$$\nabla W_{k,s}=\frac{p_{k,s}}{L_{k}}\sum_{k=1}^{K}p_{k,s}\sum_{l=1}^{L_{k,c}}\sum_{c=1}^{C}(1+\alpha_{k,s,c})X_{k,c,l}\delta_{k,s,c,l}$$
(13)$$\nabla p_{s,k}=\frac{1}{2L_{k}}\sum_{l=1}^{L_{k,c}}\sum_{c=1}^{C}(1+\alpha_{k,s,c}){\left(y_{k,c,l}-W_{k,s}X_{k,c,l}\right)}^{2}+\frac{1}{L_{k}}\sum_{k=1}^{K}p_{k,s}\sum_{l=1}^{L_{k,c}}\sum_{c=1}^{C}(1+\alpha_{k,s,c})W_{k,s}X_{k,c,l}\delta_{k,s,c,l}$$
where $\delta_{k,s,c,l}$ denotes the lth sample of cth class for kth client in the sth round of federations. It can be seen that the network model parameters of each client are updated in the direction of a decreasing prediction error during each round of federation. For the client with the better local model performance, the network model parameters are updated to a smaller extent so that the parameters are updated around the optimal values. In contrast, a larger update gradient is given to the model parameters of the client with poor local model performance so as to quickly find the optimal value to improve the model performance.

Once the federation model and the local model are both well trained by joint optimization, the online fault diagnosis can be conducted. The flowchart of the federated learning-based fault diagnosis method guided by intra-client imbalance degree is shown in Figure 5.

## 4. Experiment and Analysis

Rolling bearing is important component in the motor driven system [33]. In this section, the effectiveness of the proposed federated learning method is verified by the bearing dataset from Case Western Reserve University [34,35,36]. 

### 4.1. Experimental Data Description 

The time domain vibration data used in this paper is collected by the drive-side acceleration sensor with sampling frequency 12 kHz and motor load 1 HP with 1772 rpm speed. There are four types of faults in total: normal, inner race, outer race, and ball. Motor bearings were implanted with faults through electrical discharge machining techniques with a fault diameter of 0.021 inches. Outer race faults were located at 6 o’clock. The collected vibration signals were used to construct the dataset with a sliding window of 400 and step size of 30, and the fault labels are assigned as shown in Table 1.

### 4.2. Experimental Design

To verify the effectiveness of the proposed method, the fault diagnosis experiments shown in Table 2 are designed where both intra-client imbalance and inter-client imbalance exist. Three clients participate in the federated learning.

The test set of each client is a balanced dataset containing 1000 samples. Experiments 1–4 are designed to verify the effectiveness of the proposed method when the degree of imbalance mode mismatch between clients changes.

DNN is chosen as the network model for each client, and the number of neurons in each layer is 400/600/300/100/4, respectively. The hidden layer uses the relu activation function. The output layer uses the softmax activation function. The number of federations is 100. The exponential decay learning rate is used in this paper; the initial learning rate is 0.05, the decay steps are 50 rounds, and the decay rate is 0.98. The proposed method is compared with existing methods. The model and explanation used for the comparison are shown in Table 3.

### 4.3. Analysis of Experimental Results

It is difficult to obtain an effective fault diagnosis model when the sample size of a single client is limited and there is imbalanced data. Federated learning can achieve a joint update of multiple clients without sharing data directly. However, when there is a data imbalance intra-client, each class sample has a unique contribution degree to the federation model. Existing methods ignore the contribution of the intra-client imbalance to the federation model, affecting the effectiveness of federated learning. Therefore, this paper outlines a federated learning method that can measure the imbalance degree intra-client and guide the design of federation aggregation strategy by cost-sensitive learning. Experiments 1–4 were designed based on the mismatch degree of the imbalance mode between clients, and the experimental results are listed in Table 4, Table 5, Table 6 and Table 7.

Comparing columns 2–4 in row 2 of Table 4, it can be seen that the fault diagnosis accuracy of Client 1 is lower than Client 3, because Client 1 has less training data and has difficulty in training an effective fault diagnosis model without federated learning. Comparing rows 2 and 3 of Table 4, it can be seen that the fault diagnosis accuracy of FedAvg is higher than DNN because FedAvg can realize the joint optimization of different clients without sharing data directly. Comparing rows 3 and 4 of Table 4, it can be seen that the fault diagnosis accuracy of FedAvg-RL for Client 2 is lower than FedAvg, because the imbalance degree of federated center data on the aggregation model is not representative of the actual imbalance degree for each client when there is imbalance mode mismatch between clients. Comparing rows 3 and 5 of Table 4 with row 6, it can be seen that the diagnosis accuracy of FedCA-TDD and FA-FedAvg is higher than that of FedAvg because FedCA-TDD assigns aggregation weights for each class in the output layer based on the number of samples of each class for each client, and FA-FedAvg assigns aggregation weights based on the model performance metric F1-score for each client, showing the importance of designing an improved federation aggregation strategy to improve fault diagnosis accuracy. Comparing rows 5 and 6 of Table 4 with row 7, it can be seen that FedJuas can achieve higher fault diagnosis accuracy. This is because FedJuas constructs the federated center loss function in the federation aggregation stage that drives the joint update of the network model parameters and aggregation weights for each client, which improves the effectiveness of federation aggregation by means of learning. Comparing rows 7 and 8 of Table 4, it can be seen that Fed_ICID can achieve higher fault diagnosis accuracy than FedJuas. Because Fed_ICID can accurately measure the inter-class imbalance degree of each client as the contribution to the federation model and adding it to the federation aggregation strategy design of FedJuas, it can effectively guide the learning process of the optimal federation aggregation strategy, resulting in an average diagnostic accuracy of 96.37% for the clients. When the intra-client imbalance degree is 5:1, the fault diagnosis accuracy of each method is shown in Table 5.

Comparing Table 4 and Table 5, it can be seen that the fault diagnosis accuracy of each model decreases; this is because as the mismatch degree of imbalance mode between clients increases, the model variability of inter-client becomes greater, affecting the effectiveness of federation aggregation. In addition, the increased degree of inter-class imbalance for intra-client poses difficulties for model optimization. However, by measuring the degree of inter-class imbalance for each client to guide the design of the federation aggregation strategy, Fed_ICID can guarantee the effectiveness of federation aggregation, and therefore the fault diagnosis accuracy is 22.54% higher than FA-FedAvg. Comparing rows 2 and 3 in Table 5, it can be seen that the fault diagnosis accuracy of FedAvg is higher than DNN because FedAvg can achieve joint optimization of multiple client models. Comparing rows 3 and 4 in Table 5, it can be seen that the fault diagnosis accuracy of FedAvg-RL for Client 1 and Client 2 is lower than FedAvg, due to the fact that the federated center assigns the same degree of inter-class imbalance to each client, which does not accurately measure the unique degree of imbalance for each client and therefore affects the fault diagnosis effectiveness. Comparing rows 5 to 7 in Table 5, it can be seen that FedJuas achieves higher fault diagnosis accuracy due to the fact that FedCA-TDD and FA-FedAvg determine the aggregation weights for each client based on the number of class samples and the model performance metrics, respectively, which may not be optimal aggregation weights. However, FedJuas constructs the loss values of the federated center by aggregating the loss values of each client to achieve joint optimization of the aggregation weights and model parameters for each client, which improves the effectiveness of the federation aggregation strategy, so each client can better benefit from the federated center and improve the fault diagnosis accuracy. Comparing rows 7 and 8 in Table 5, it can be found that Fed_ICID can obtain higher fault diagnosis accuracy than FedJuas, due to the fact that Fed_ICID can measure the imbalance degree of intra-client and is used to improve the federation aggregation strategy of FedJuas when there is imbalance mode mismatch between clients. When the intra-client imbalance degree is 7:1, the fault diagnosis accuracy of each method is shown in Table 6.

Comparing Table 4, Table 5 and Table 6, it can be seen that the fault diagnosis accuracy of each method decreases as the mismatch degree of imbalance mode between clients increases. Comparing rows 2 and 3 in Table 6, it can be seen that FedAvg has a higher fault diagnosis accuracy than DNN due to the joint optimization of different client models achieved by federated learning without sharing data directly. Comparing rows 3 and 4 in Table 6, it can be seen that the fault diagnosis accuracy of FedAvg-RL for Client 1 is lower than FedAvg, due to the fact that the degree of data imbalance in the federated center does not accurately measure the unique imbalance of each client, and therefore affects the fault diagnosis accuracy. Comparing rows 3 to 6 and 8 of Table 6, it can be seen that Fed_ICID can achieve higher fault diagnosis accuracy because Fed_ICID can accurately measure the imbalance degree of each client and construct the federation aggregation strategy by the cost-sensitive loss function of each client, learn the aggregation weights and further update the model parameters of each client, different from other methods where the federation aggregation strategy is determined based on sample size or model performance metrics of each client, thus each client can maximum benefit from the federation aggregation model. When the intra-client imbalance degree is 23:1, the fault diagnosis accuracy of each method is shown in Table 7.

Comparing Table 4, Table 5, Table 6 and Table 7, it can be shown that the fault diagnosis accuracy of each method decreases as the mismatch degree of imbalance mode between clients increases, indicating that the greater the degree of data imbalance between clients, the greater the impact on the fault diagnosis results. Comparing row 8 of Table 6 with row 8 of Table 7, it can be seen that the fault diagnosis method of Fed_ICID can achieve an average fault diagnosis accuracy of 93.03% at an intra-client imbalance degree of 23:1 due to its ability to measure the degree of inter-class imbalance for inter-client and construct the federated center loss function through the cost-sensitive loss function of each client that drives the joint update of model parameters and aggregation weights. Comparing rows 6 and 8 of Table 7, it can be seen that the diagnosis accuracy of Fed_ICID is 23.33% higher than FA-FedAvg, which indicates that measuring the inter-class imbalance degree for each client and used in the design of federation aggregation strategy can effectively improves the fault diagnosis accuracy of federated learning, which cannot be done by the existing federation aggregation strategy based on the sample size or model performance metrics, and is the advantage of the proposed method in this paper.

To further present the diagnosis effect of the proposed method, the confusion matrix of each model’s fault diagnosis result when the intra-client imbalance degree is 23:1 is shown in Figure 6.

Each subfigure in Figure 6 includes three confusion matrixes, representing the fault diagnosis results for each of the three clients, where the number of samples with correct diagnosis results is represented by the diagonal numbers. The darker color indicates that more samples are correctly diagnosed for each type of data. Comparing the subfigures (a,b), it can be seen that the number of samples correctly diagnosed by federated learning is higher than that of DNN, which is due to the fact that federated learning can achieve joint optimization of multiple clients while ensuring the data privacy of each client. Comparing subfigures (b,e), it can be seen that the number of samples correctly diagnosed by FA-FedAvg based on model metrics F1-score is higher than that of FedAvg. This is because FA-FedAvg can assign dynamic aggregation weights for each client based on the model performance in each round of federation. Comparing subfigures (d,e,g), it can be seen that Fed_ICID can accurately measure the unique inter-class imbalance degree of each client, whereby it is used as the contribution degree of each client to the federation model, and is used for the construction of the federation aggregation strategy that includes cost-sensitive learning, so the misclassified fault samples are the least and the fault diagnosis is the best. Figure 7 shows the histogram of average fault diagnosis accuracy for Experiments 1–4.

## 5. Conclusions and Future Work

Federated learning can realize the joint optimization of different client models by the federation aggregation strategy without sharing data directly. The existing federation aggregation strategy tends to reflect the imbalance degree of inter-client based on the sample size or the model performance metrics of each client to obtain federation aggregation weights. However, it only considers the whole model performance of each client rather than how the inter-class imbalance of the intra-client will affect the effectiveness of the federation aggregation model for the case when there is imbalance mode mismatch between clients. Therefore, it is significant to design a federated learning framework driven by intra-client imbalance degree to guide the construction of federation aggregation strategy for the case when there is an imbalance mode mismatch between clients.

The degree of inter-class imbalance for each client can be determined by the gain of each class sample of imbalanced data when updated separately on the federation model of balanced samples. By aggregating the cost-sensitive loss function computed by inter-class imbalance degree to obtain the federated center loss function, which is used to drive the joint updating of local model parameters and federation model parameters so that each client can benefit maximally from the federation aggregation model. Experimental results show that the average fault diagnosis accuracy of the proposed Fed_ICID method is 23.33% higher than existing method when there is significant mismatch of imbalance mode between clients.

This study only focuses on solving the problem of labeled samples with imbalance without considering the massive amounts of unlabeled data. So, the question of how to design a semi-supervised federated learning method using a combination of labeled and unlabeled data to solve the problem of sample imbalance is a promising research direction.

## Figures and Tables

**Figure 1 entropy-25-00606-f001:**
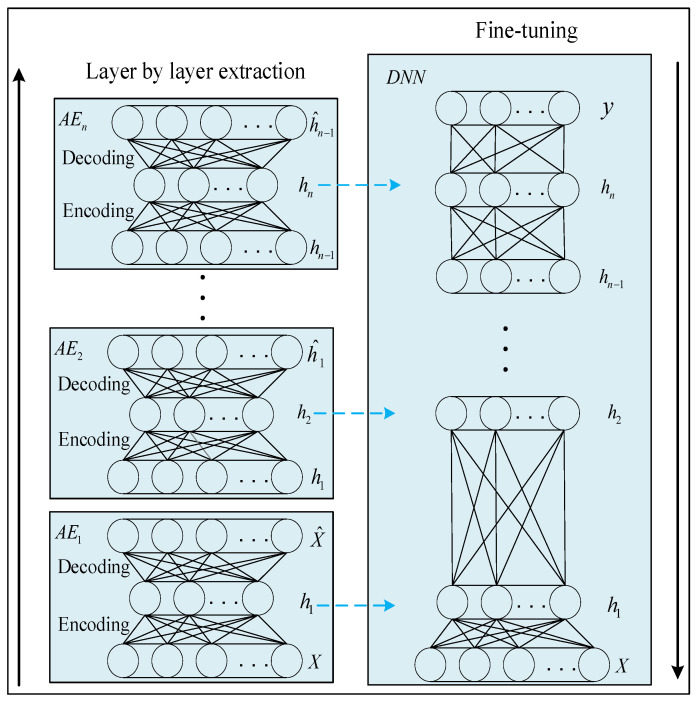
Network structure diagram of DNN.

**Figure 2 entropy-25-00606-f002:**
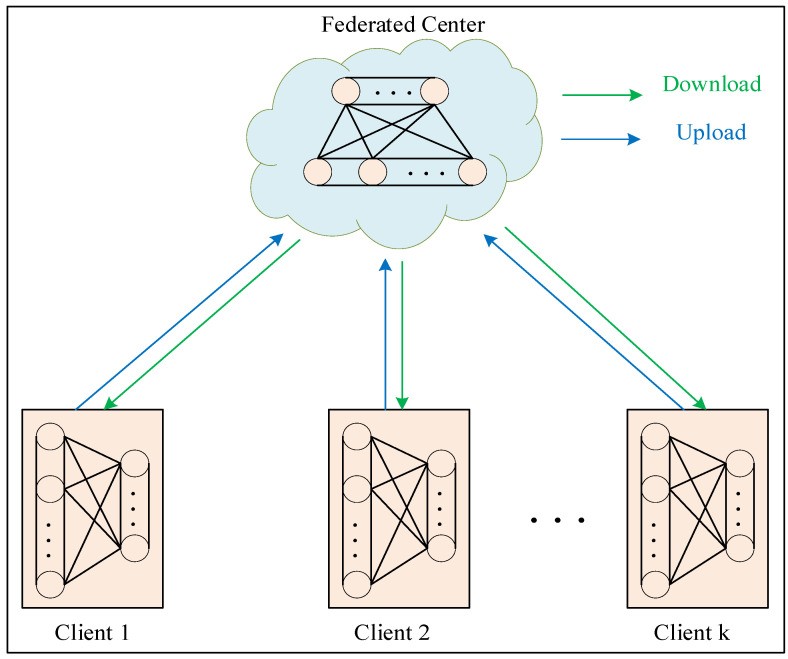
Federated learning schematic.

**Figure 3 entropy-25-00606-f003:**
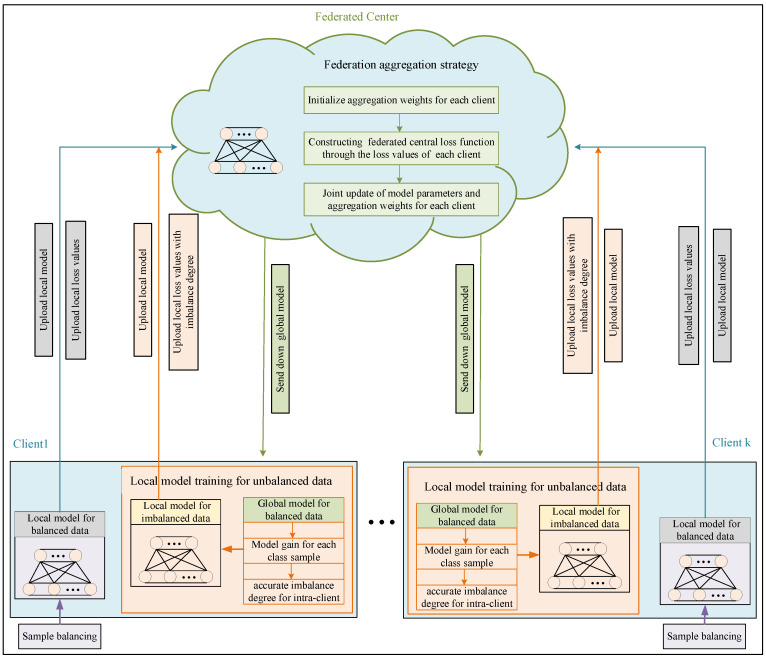
A federated learning framework driven by intra-client imbalance degree.

**Figure 4 entropy-25-00606-f004:**
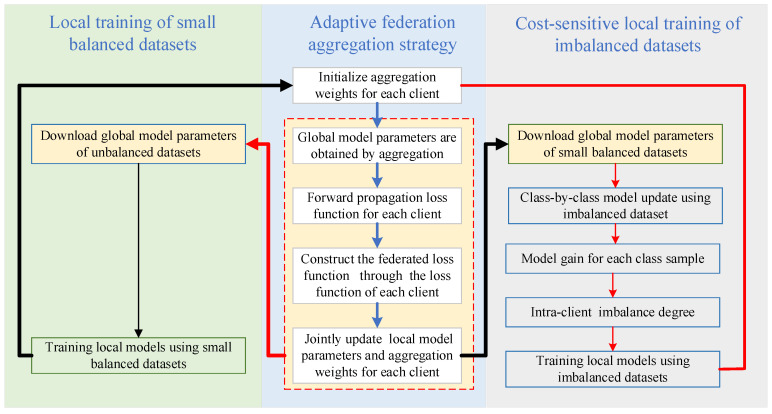
Block diagram of the federated learning method driven by intra-client imbalance degree.

**Figure 5 entropy-25-00606-f005:**
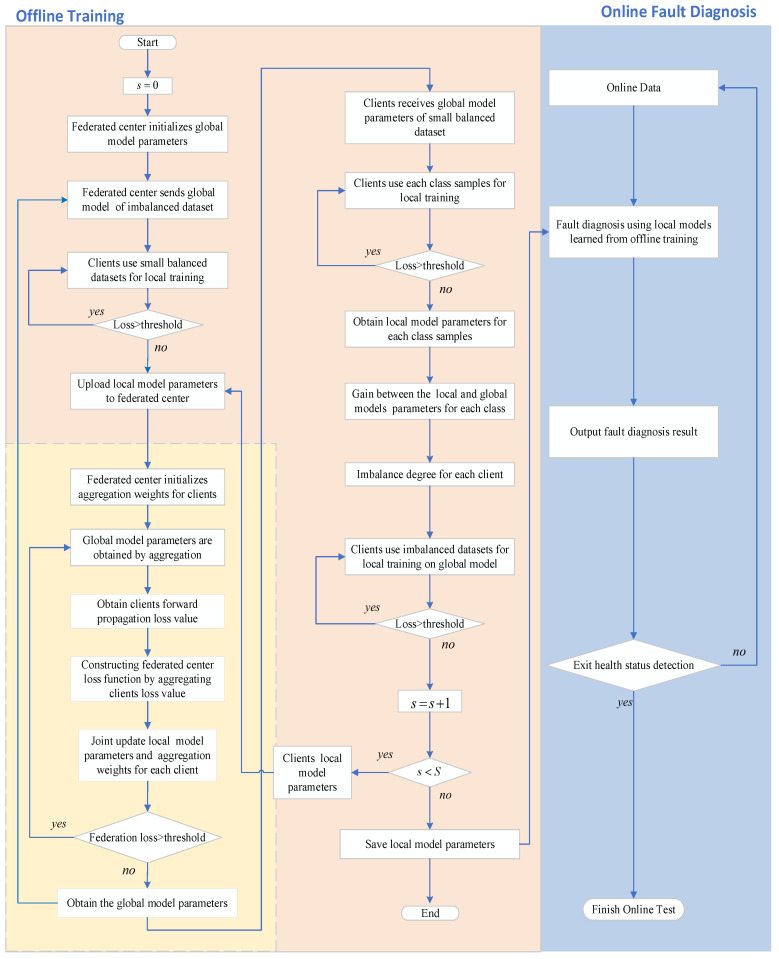
The flowchart of the federated learning-based fault diagnosis method guided by intra-client imbalance degree.

**Figure 6 entropy-25-00606-f006:**
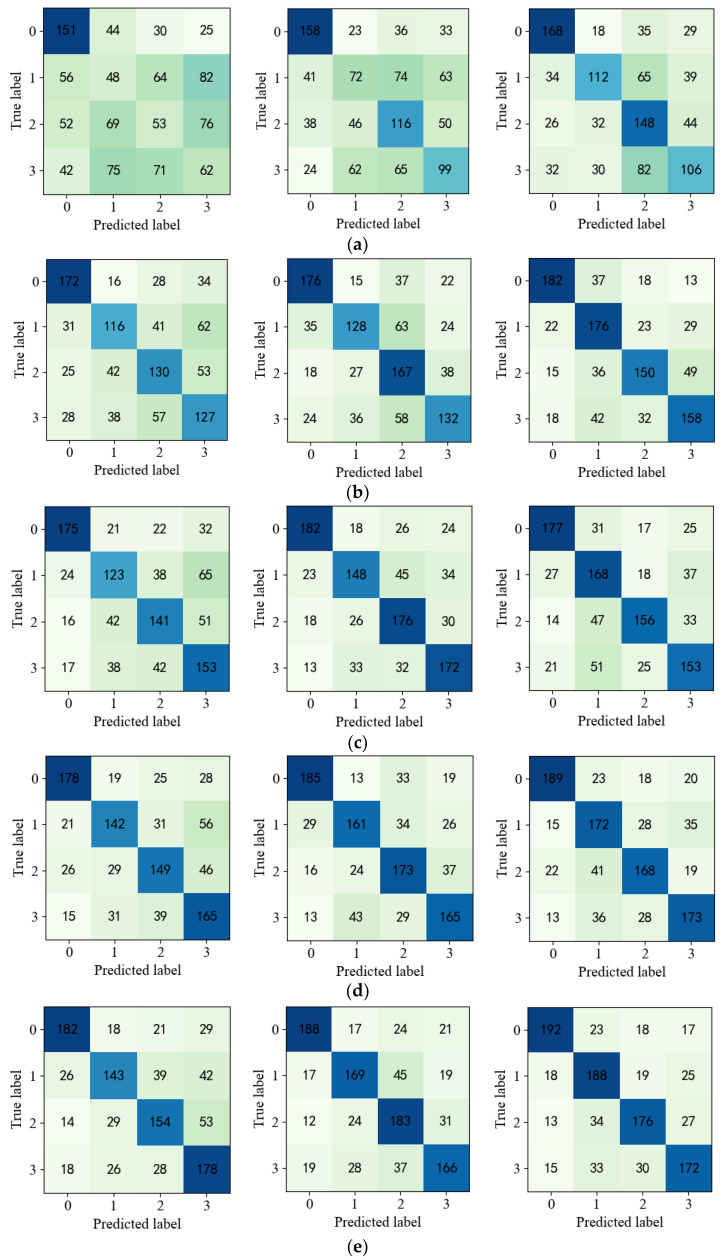
Confusion matrix for each method’s fault diagnosis result when the intra-client imbalance degree is 23:1. (**a**) DNN. (**b**) FedAvg. (**c**) FedAvg-RL. (**d**) FedCA-TDD. (**e**) FA-FedAvg. (**f**) FedJuas. (**g**) Fed_ICID.

**Figure 7 entropy-25-00606-f007:**
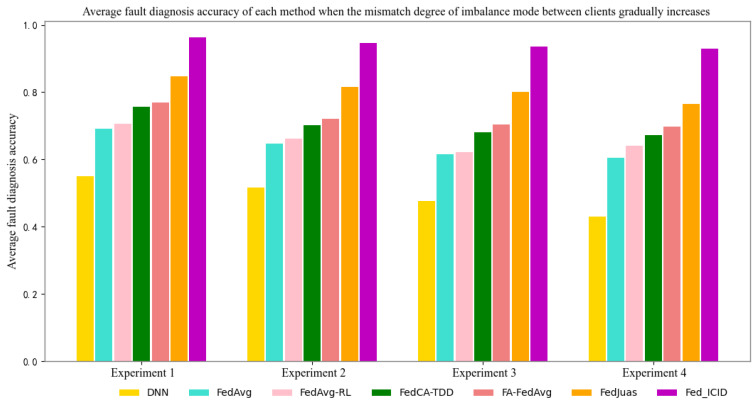
Histogram of the results of experiments 1–4.

**Table 1 entropy-25-00606-t001:** Fault types.

Fault Type	Fault Diameter	Fault Label	Load (HP)	Speed (rpm)
Normal	0	0	1	1772
Inner race	0.021	1	1	1772
Outer race	0.021	2	1	1772
Ball	0.021	3	1	1772

**Table 2 entropy-25-00606-t002:** Experimental design.

Experiment	Intra-Client Imbalance Degree	Inter-Class Sample Size of Clients	Inter-Client Training Sample Size	Test Set Sample Size	Intra-Client Majority Classes/Minority Classes
Experiment 1	2:1	16/8/8/16	48/192/384	1000	03/12
		64/32/64/32			02/13
		128/128/64/64			01/23
Experiment 2	5:1	20/4/4/20	48/192/384	1000	03/12
		80/16/80/16			02/13
		160/160/32/32			01/23
Experiment 3	7:1	21/3/3/21	48/192/384	1000	03/12
		84/12/84/12			02/13
		168/168/24/24			01/23
Experiment 4	23:1	23/1/1/23	48/192/384	1000	03/12
		92/4/92/4			02/13
		184/184/8/8			01/23

**Table 3 entropy-25-00606-t003:** Relevant experimental model used for comparison.

Model	Model Explanation
DNN	Traditional deep learning deep neural networks without federated learning
FedAvg [26] FedAvg-RL [30]	Traditional federal average aggregation strategy based on sample size Federated averaging method with local model ratio loss
FedCA-TDD [27]	Class-weighted aggregation strategy based on class sample size
FA-FedAvg [28]	Improved federated aggregation strategy based on model metrics F1-score
FedJuas	Federated strategy of joint update aggregation weights and model parameters proposed in this paper but without cost-sensitive learning
Fed_ICID	An inter-client federated learning method based on accurately measure the intra-client imbalance degree proposed in this paper

**Table 4 entropy-25-00606-t004:** Fault diagnosis accuracy when the intra-client imbalance degree is 2:1.

Model	Client 1	Client 2	Client 3	Mean
DNN	45.3%	54.8%	65.2%	55.10%
FedAvg	63.2%	68.5%	75.8%	69.17%
FedAvg-RL	65.6%	66.9%	79.7%	70.73%
FedCA-TDD	70.5%	73.2%	83.4%	75.70%
FA-FedAvg	72.4%	75.6%	82.8%	76.93%
FedJuas	83.7%	84.3%	86.5%	84.83%
Fed_ICID	95.5%	96.7%	96.9%	96.37%

**Table 5 entropy-25-00606-t005:** Fault diagnosis accuracy when the intra-client imbalance degree is 5:1.

Model	Client 1	Client 2	Client 3	Mean
DNN	42.1%	51.3%	61.9%	51.77%
FedAvg	61.4%	64.3%	68.8%	64.83%
FedAvg-RL	58.7%	63.7%	75.9%	66.10%
FedCA-TDD	64.8%	68.9%	76.7%	70.13%
FA-FedAvg	67.3%	69.3%	79.5%	72.03%
FedJuas	79.8%	81.4%	83.7%	81.63%
Fed_ICID	93.8%	94.2%	95.7%	94.57%

**Table 6 entropy-25-00606-t006:** Fault diagnosis accuracy when the intra-client imbalance degree is 7:1.

Model	Client 1	Client 2	Client 3	Mean
DNN	38.7%	46.7%	57.5%	47.63%
FedAvg	60.3%	61.6%	62.7%	61.53%
FedAvg-RL	57.1%	63.3%	66.5%	62.30%
FedCA-TDD	65.6%	68.8%	70.2%	68.20%
FA-FedAvg	69.2%	69.6%	72.5%	70.43%
FedJuas	80.6%	81.3%	78.3%	80.07%
Fed_ICID	92.2%	94.5%	93.9%	93.53%

**Table 7 entropy-25-00606-t007:** Fault diagnosis accuracy when the intra-client imbalance degree is 23:1.

Model	Client 1	Client 2	Client 3	Mean
DNN	31.4%	44.5%	53.4%	43.10%
FedAvg	54.5%	60.3%	66.6%	60.47%
FedAvg-RL	59.2%	67.8%	65.4%	64.13%
FedCA-TDD	63.4%	68.4%	70.2%	67.33%
FA-FedAvg	65.7%	70.6%	72.8%	69.70%
FedJuas	70.6%	78.7%	80.3%	76.53%
Fed_ICID	91.5%	93.4%	94.2%	93.03%

## Data Availability

The data involved in this article have been presented in the article.

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
