# Peer review of "Federated Learning Based Fault Diagnosis Driven by Intra-Client Imbalance Degree"

_entropy, 2023, doi:10.3390/e25040606_

Round 1

Reviewer 1 Report

The following observations are listed below,

1. For the abstract, overall findings and conclusions must be stated. 

2. Title of the work is not exactly matching with presented content and not clear. 

3. What is the objective of the research?. Have you discussed in proposed method and results section?

4. Results and obtained values details are not shown in abstract. Why the research work is important?

5. Ambiguities in section 2 and 3 because of difficult to identify which one is proposed methodology?

6. Improper reference citations and styles used in this paper. Refer the mdpi author guidelines for further details. 

7. How the intra- and inter-client imbalance driven approach used in this  per?. Directly coined this term without justifying the use of approach. 

8. In Fig 8, what is the X-axis and Y-axis units? Simply the authors used numerical values and plotted the graph. 

9. What is the data source of generated graph/charts?. It is synthetic data samples. How do you used this data?

Author Response

Ref: entropy-2271633

Title: Federated learning based fault diagnosis driven by intra-client imbalance degree

Authors: Funa Zhou*, Yi Yang, Chaoge Wang, Xiong Hu

Dear Editors-in-Chief and Reviewers,

We deeply appreciate the time and effort you have spent in reviewing our manuscript. Thank you very much for giving us a chance to respond to the Reviewers’ comments and we would like to thank the Reviewers for their constructive comments on our manuscript. We have considered the comments of each Reviewer and made some changes and corrections in the manuscript accordingly. Our changes and response are presented in the following. The point-to-point answers and explanations for all comments were listed following this letter, and the modified words and sentences are marked with blue. We hope that the Editors and Reviewers will be satisfied with the revisions for the original manuscript. If you have any question about this paper, please contact us without hesitate.

Thanks and Best regards!

                                                                Yours Sincerely,

Funa Zhou*, Yi Yang, Chaoge Wang, Xiong Hu

Corresponding author: Funa Zhou

Department of Electrical Engineering

Shanghai Maritime University

No. 1550, Linggang avenue, Pudong Dsitrict, Shanghai

Author Response

(The authors gave the same response as above.)

Reviewer 3 Report

1. Proper flowchart is required.  There are missed “yes” or “no” for decision functions in Fig.5.

2. More clear confusion matrix in fig.8 is suggested. Fig.6 [32] may be not required.

3. Proper units are suggested.  Such as  12Khz  => 12 kHz in line 426

4. All symbols in equation should be clearly defined for easy read.

5.  In section 4 (Table 1), more detailed of CWRU dataset are suggested such as testing speed (rpm) for inner race, outer race, and ball fault.     

6.  More detailed descriptions of compared methods as shown in Table 3 are suggested.

7. More references about Deep learning for Rotating Machines or Rolling Bearing are suggested.  Such as

(1)        Smith, W.A.; Randall, R.B. Rolling Element Bearing Diagnostics Using the Case Western Reserve University Data: A Benchmark Study.  Mech. Syst. Signal Processing 2015, 6465, 100131. https://doi.org/10.1016/j.ymssp.2015.04.021 .   (CWRU )

(2)        Huang, H.; Baddour, N. Bearing Vibration Data Collected under Time-Varying Rotational Speed Conditions. Data Brief 2018, 21, 17451749. https://doi.org/10.1016/j.dib.2018.11.019 .  (University of Ottawa in Canada)

(3)        Jacob Hendriks, Patrick Dumond, D.A. Knox, Towards better benchmarking using the CWRU bearing fault dataset   https://www.sciencedirect.com/science/article/pii/S0888327021010499  

8. The quantity results and main contribution are suggested in conclusions.

9. The manuscript seems too wordy.  A concise version is suggested.

Author Response

(The authors gave the same response as above.)

Reviewer 4 Report

To the authors

1. Fig.4

Both "Local model parameters are obtained through the small barance dataset" and "local model parameters are obtained through the imbalance dataset" are sent into the Federated Center.

Are they combined on the way to Federated Center? 

I thik it is a little confusing.

2. Equ.(2)

What does "r" mean?

3. Section 4

This paper is lack of explanation about analysis overall.

Induced detailed numerical values (such as global model parameter, local model parameters, sample weight alpha, etc.) are not presented in this paper at all.

Derived these detailed values have to be presented and be explained more.

4. Section 4.2

What kind of vibration data are they used in the comparison of sevral methods.

Time series data or frequency spectrum?

The explanation is insufficient.

5. line 419

"Outer race faults were classified as 3 o'clock, 6 o'clock, and 12 o'clock."

The bearing is circlar shape object. What does these values mean? 

6. line 426 "12Khz"

Please check if the notation "12Khz" is correct. "12kHz"?

7. Table 2 

Inter-client sample size ratio is 50:200:400.

What does the ratio mean?

The description is lacking.

8.

line 451 "400/300/600/100/4"

Since the number of Label is 4 in the Table1, the number of neurons of output layer seems 4. The number of neurons of input layer is 400?

How should we understand this represensation?

9.

line 567, 573 "Table 8"

Table 8 is correct? Table7?

10. Table 4

The model named "Cmdd-FedJuas" seems to appear suddenly.

But after reading, I found that row 7 and 8 seemed to be the proposal methods in line 499.

To make it easy to understand, you should explain them as proposed methods first.

11. line 582

Is Fig.9 a histogram?

By comparison between Fign9 and Table4-7, Fig.9 seems to be results of "mean value" of  the accuracy. 

If my guess is right, please explain it correctly.

Author Response

(The authors gave the same response as above.)

Round 2

Reviewer 1 Report

All the comments are addressed in this revised paper. No more queries. 

Reviewer 4 Report

I don't think I have any questions or concerns.